# WHERE PRIOR LEARNING CAN AND CAN'T WORK IN UNSUPERVISED INVERSE PROBLEMS

## ABSTRACT

Linear inverse problems consist in recovering a signal from its noisy observation in a lower dimensional space. Many popular resolution methods rely on data-driven algorithms that learn a prior from pairs of signals and observations to overcome the loss of information. However, these approaches are difficult, if not impossible, to adapt to unsupervised contexts – where no ground truth data are available – due to the need for learning from clean signals. This paper studies situations that do or do not allow learning a prior in unsupervised inverse problems. First, we focus on dictionary learning and point out that recovering the dictionary is unfeasible without constraints when the signal is observed through only one measurement operator. It can, however, be learned with multiple operators, given that they are diverse enough to span the whole signal space. Then, we study methods where weak priors are made available either through optimization constraints or deep learning architectures. We empirically emphasize that they perform better than hand-crafted priors only if they are adapted to the inverse problem.

## 1 INTRODUCTION

Linear inverse problems are ubiquitous in observational science such as imaging (Ribes & Schmitt, 2008), neurosciences (Gramfort et al., 2012) or astrophysics (Starck, 2016). They consist in reconstructing signals $X \in \mathbb{R}^{n \times N}$ from remote and noisy measurements $Y \in \mathbb{R}^{m \times N}$ which are obtained as a linear transformation $A \in \mathbb{R}^{m \times n}$ of $X$, corrupted with noise $B \in \mathbb{R}^{m \times N}$: $Y = AX + B$. As the dimension $m$ of $Y$ is usually much smaller than the dimension $n$ of $X$, these problems are ill-posed, and several solutions could lead to a given set of observations. The uncertainty of the measurements, which can be noisy, increases the number of potential solutions. Therefore, practitioners rely on prior knowledge of the data to select a plausible solution among all possible ones.

On the one hand, hand-crafted priors relying on sparsity in a basis produce satisfying results on specific data, such as wavelets in imaging or Gaborlets in audio (Mallat, 2008). However, the complexity and variability of the signals often make *ad hoc* priors inadequate. On the other hand, the prior can be learned from ground truth data when available. For instance, frameworks based on Plug-and-Play (Brifman et al., 2016) and Deep Learning (Chan et al., 2016; Romano et al., 2017; Rick Chang et al., 2017) propose to integrate a pre-trained denoiser in an iterative algorithm to solve the problem. Supervised methods leveraging sparsity also allow to summarize the structure of the signal (Elad, 2010). In particular, dictionary learning (Olshausen & Field, 1997; Aharon et al., 2006; Mairal et al., 2009) is efficient on pattern learning tasks such as blood cell detection or MEG signals analysis (Yellin et al., 2017; Dupré la Tour et al., 2018). Nevertheless, these methods require clean data, sometimes available in audio and imaging but not in fields like neuroimaging or astrophysics.

While data-driven methods have been extensively studied in the context of supervised inverse problems, recent works have focused on unsupervised scenarios and provided new algorithms to learn from corrupted data only (Lehtinen et al., 2018; Bora et al., 2018; Liu et al., 2020). Chen et al. (2021) and Tachella et al. (2022) demonstrate that a necessary condition to learn extensive priors from degraded signals is either to measure them with multiple operators which span the whole space, or to introduce weak prior knowledge such as group structures and equivariance in the model when only one operator is available. Other works based on Deep Learning have leveraged successful architectures to recover images without access to any ground truth data. In particular, Deep Image Prior shows that CNNs contain enough prior information to recover an image in several

inverse problems, such as denoising or inpainting (Ulyanov et al., 2018). Finally, a few works have demonstrated that it is possible to learn dictionaries from incomplete data, especially in the context of missing values or inpainting in imaging (Szabó et al., 2011; Studer & Baraniuk, 2012; Naumova & Schnass, 2017). Another line of work studied online factorization of large matrices by aggregating partial information randomly selected from the data at each iteration (Mensch et al., 2016; 2017). This is equivalent to learning a dictionary from incomplete data, except that one sample can be looked at multiple times from different angles, which is hardly possible in an inverse problem context.

**Contributions**   In this paper, we demonstrate practical limitations of prior learning methods for unsupervised inverse problems. We first provide an analysis of dictionary learning when the data is measured with a single or multiple operators. As mentioned by Tachella et al. (2022), "seeing the whole space" is a necessary condition to learn a good prior from the data, as nothing can be recovered in the kernel of the operator $A$. However, we point out that this is not sufficient in the case of dictionary learning. Indeed, the problem is made harder by the measurement operators, and is sometimes unfeasible even with access to the whole space. Then we study the practical behavior of methods heavily relying on convolutions in cases where they work well (inpainting) and in cases where they fail because the prior is too weak (deblurring), and provide experiments complementary to the theoretical study of Tachella et al. (2022). We present three examples, namely Convolutional Dictionary Learning, Deep Image Prior, and Plug and Play, and train the prior "as is" in the range space without relying on any data augmentation technique or equivariance. Finally, we show that the difficulty is deeper than the unsupervised setting by studying what happens in a self-supervised setting when training on ground truth data. In particular, we emphasize that stronger prior information is necessary to link low and high frequencies in deblurring, even in this simpler context.

## 2   THE MAIN BOTTLENECK OF PRIOR LEARNING IN INVERSE PROBLEMS

For inverse problems, the dimension of the measurements $m$ is often smaller than the dimension of the signal $n$. This dimension reduction implies that information on the signal contained in the null space of $A \in \mathbb{R}^{m \times n}$ is lost during the observation process, and needs to be reconstructed from the observed signal. We first aim to study the impact of this degradation on constraint-free prior learning through the lens of dictionary learning.

### 2.1   DICTIONARY LEARNING WITH A SINGLE MEASUREMENT OPERATOR.

Dictionary learning assumes that the signal can be decomposed into a sparse representation in a redundant basis of patterns – also called atoms. In other words, the goal is to recover the signals $X \in \mathbb{R}^{n \times N}$ as $DZ$ where $Z \in \mathbb{R}^{L \times N}$ are sparse codes and $D \in \mathbb{R}^{n \times L}$ is a dictionary. Taking the example of Lasso-based dictionary learning, recovering $X$ would require solving a problem of the form

$$\min_{Z \in \mathbb{R}^{L \times N}, D \in \mathcal{C}} \frac{1}{2} \|ADZ - Y\|_2^2 + \lambda \|Z\|_1 \quad , \tag{1}$$

where $\lambda$ is a regularization hyperparameter and $\mathcal{C}$ is a set of constraints, typically set so that columns of $D$ have norm smaller than 1. We first aim to see the impact of $A$ on the algorithm ability to recover a proper dictionary. In Proposition 2.1, we focus on inpainting where the measurement operator is a binary mask or equivalently a diagonal matrix with $m$ non-zeros elements.

**Proposition 2.1.** *Let $A = diag(\lambda_1, \cdots, \lambda_n) \in \mathbb{R}^{n \times n}$ be a diagonal measurement matrix where $m < n$, $\lambda_1 \geq \cdots \geq \lambda_m > 0$ and $\lambda_{m+1} = \cdots = \lambda_n = 0$. Let $D_0 \in \mathbb{R}^{n \times L}$ and $D'$ be such that*

$$D' = \begin{pmatrix} \frac{\|D_{0,j}\|}{\|D_{0,j,m}\|} D_{0,j,m} \\ 0_{n-m} \end{pmatrix}_{1 \leq j \leq L}, \; where \; D_0 = \begin{pmatrix} D_{0,m} \\ D_{0,n-m} \end{pmatrix}$$

*Then*

$$\min_Z \frac{1}{2} \|AD'Z - Y\|_2^2 + \lambda \|Z\|_1 \leq \min_Z \frac{1}{2} \|AD_0Z - Y\|_2^2 + \lambda \|Z\|_1 \; .$$

All proofs are deferred to Appendix C. In this simple case, our proposition shows that the optimal dictionary must be 0 in the null space of $A$. The core idea behind the proof is that due to invariances, the optimal solution for dictionary learning is contained in an equivalence class $\{PSD' + V\}$ where

$P$ is a permutation matrix, $S$ is a scaling matrix, $D'$ is a matrix of rank $m$ and $V$ is a matrix of rank $n - m$ such that $PSD' \in \ker(A)^{\perp}$ and $V \in \ker(A)$. Given a dictionary $PSD' + V$ in this equivalence class, the dictionary $PSD'$ is always a better minimizer after proper rescaling. Therefore, the solver puts to 0 all directions from which $A$ loses the information to maximize the input from the others. Proposition 2.2 generalizes Proposition 2.1 to the case of rectangular matrices.

**Proposition 2.2.** *Let $A \in \mathbb{R}^{m \times n}$ be a measurement matrix where $m < n$, and let $Y \in \mathbb{R}^{m \times N}$ be the observed data. If a dictionary $D \in \mathbb{R}^{n \times L}$ minimizes $\min_{Z \in \mathbb{R}^{L \times N}, D \in \mathcal{C}} \frac{1}{2} \| ADZ - Y \|_2^2 + \lambda \| Z \|_1$, then $D \in \ker(A)^{\perp}$.*

Similarly to what happens in inpainting, nothing can be recovered in the null space of $A$. Thus, we can only expect to learn a dictionary of rank $m$ with a single measurement matrix. This result relates to the one from Tachella et al. (2022) which says that the signals cannot be recovered where there is no information.

**Dimension reduction makes dictionary learning harder in the range space.** Even in the range space of the signal, a good dictionary cannot always be learned reliably. Guarantees of identifiability of the dictionary or local recovery are strongly based on the ability of the sparse coding algorithm to recover an accurate estimation of $Z$ (Arora et al., 2015; Gribonval et al., 2015; Chatterji & Bartlett, 2017). As the dimension of the measurement $m$ becomes smaller than the dimension of the signal $n$, these conditions are not valid anymore. As an example, if $D$ is a Gaussian random dictionary, the theory of compressed sensing states that $n \geq 2s \ln(\frac{L}{s})$ where $s$ is the sparsity of $Z$ is a sufficient condition to be able to recover $Z$ with high probability (Foucart & Rauhut, 2013). When the dictionary is degraded by a matrix $A$, this constraint becomes $m \geq 2s \ln(\frac{L}{s})$ and the sparsity level $s$ has to decrease by a ratio close to $\frac{m}{n}$ to compensate for the loss of information. This implies that recovering the part of the dictionary not contained in the null space of $A$ also becomes harder with the corruption of the data.

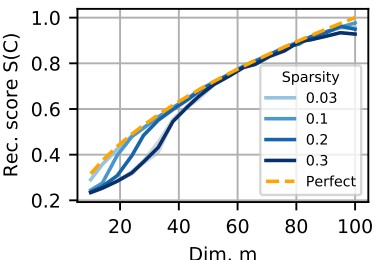

Figure 1: Recovery score for Gaussian dictionaries $100 \times 100$, after degradation by a single compressed sensing operator $m \times 100$. When the dimension $m$ decreases, the part of the dictionary not contained in the null space can be recovered only with sparse signals.

Fig. 1 shows the recovery score for data generated with a $100 \times 100$ random Gaussian dictionary, depending on the size of the measurements and on the sparsity of a Bernoulli Gaussian signal, after degradation by a single compressed sensing operator. We compare it to the perfect score that we can achieve in the range space of the operator. We evaluate the quality of the dictionary, based on the Pearson correlation of their columns. To make the metric sign and permutation invariant, we use a best linear sum assignment $S(C) = \max_{\sigma \in \mathfrak{S}_n} \frac{1}{n} \sum_{i=1}^{n} |C_{\sigma(i),i}|$, where $\mathfrak{S}_n$ is the group of permutations of $[1, n]$ and $C$ is the cost matrix whose entry $i, j$ compares the atom $i$ of the first dictionary and $j$ of the second. It is equal to 1 when the dictionary is perfectly recovered. The recovery score drops when the dimension $m$ decreases and small values of $m$ require a high sparsity level to recover the dictionary in the range of $A$.

## 2.2 SEEING THE DATA THROUGH MULTIPLE OPERATORS

Even though it is not possible to recover the whole dictionary from a single measurement operator, the situation changes when the measurement matrix is sample dependent. Indeed, several operators may span different parts of the signal space and make it possible to recover the missing part of the signal. In this section, we focus on cases where the data are observed through a set of $N_m$ measurement matrices $(A_i)_{1 \leq i \leq N_m}$, and consider the task of learning a dictionary with the associated lasso-based (Tibshirani, 1996) optimization problem

$$\min_{D \in \mathcal{C}} F(Z_A(D), D) \triangleq \sum_{i=1}^{N_m} \frac{1}{2} \| A_i D Z_{A_i}(D) - Y_i \|_2^2 + \lambda_i \| Z_{A_i}(D) \|_1 \quad,$$

$$\text{with} \quad Z_{A_i}(D) = \arg\min_{Z} \frac{1}{2} \| A_i D Z - Y_i \|_2^2 + \lambda_i \| Z \|_1 \quad.$$

(2)

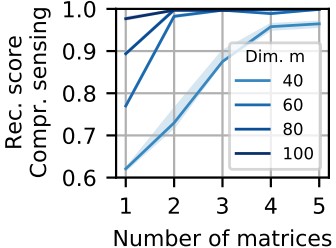 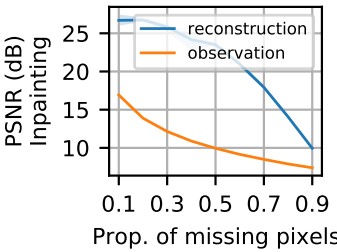 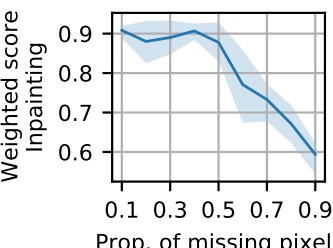

Figure 2: **(left)** Recovery score for Gaussian dictionaries $100 \times 100$, after degradation by $N_m$ compressed sensing operators $m \times 100$. $N_m \geq \lfloor \frac{n}{m} \rfloor + 1$ is necessary, but not sufficient to recover the dictionary when m is too small. In the case of inpainting, PSNR **(center)** and weighted recovery score **(right)** depend on the proportion of missing values in dictionary learning on patches from a natural image. When the dimension of the measurement space is large enough, the algorithm recovers the image and the supervised dictionary successfully.

Here $Z_A(D) = (Z_{A_1}(D), \dots Z_{A_{N_m}}(D))$ denotes the sparse codes related to each operator. This problem is non-convex and is usually solved through gradient descent, in order to find a local minimum. In the following, we study the cases when the local minima of Eq. (2) are also local minima for the problem without observation operators and provide an empirical analysis in different scenarios. With multiple measurement operators, the gradient of Eq. (2) is given by

$$\nabla_D F(Z_A(D), D) = \sum_{i=1}^{N_m} A_i^T (A_i D Z_{A_i}(D) - Y_i) Z_{A_i}(D)^T \ . \tag{3}$$

The main difficulty in studying this quantity is that the sparse codes estimate $Z_{A_i}(D)$ depends on $D$ and $A_i$. Each operator provides measurements from a limited number of samples in the data-set, and the sparse codes are different with and without $A$. Thus, we consider the simplest case where $Z = Z_{A_i}(D)$ is the same for all $A_i$, including $A = I$. This is an easier problem than the general formulation in Eq. (2). If this is not feasible, then Eq. (2) is not feasible. In this case, we have

$$\nabla_D F(Z_A(D), D) = (\sum_i A_i^T A_i) \nabla_D F(Z_I(D), D) \ . \tag{4}$$

KKT conditions imply that the gradient $\nabla_D F(Z_A(D), D)$ must vanish at local minima. Whenever $\sum_i A_i^T A_i$ is injective, $\nabla_D F(Z_I(D), D)$ vanishes if and only if $\nabla_D F(Z_A(D), D)$ vanishes. Thus, local minima of Eq. (2) are also local minima for the original problem where $A_i = I$. This means that when $\sum_i A_i^T A_i$ spans the entire space, the dictionary from the original problem can be recovered. Otherwise local minima of $F(Z_A(D), D)$ are not necessarily local minima of $F(Z_I(D), D)$. This case boils down to the case previously studied in Section 2, as $\sum_i A_i^T A_i$ is full rank whenever the rank of the matrix obtained by stacking the operators $(A_1^T, \cdots, A_{N_m}^T)$ is equal to $n$. The message from these results is essentially the same as the one from Chen et al. (2021): a necessary condition for recovery is that the operators span the whole space. It is however important to note that this is only a necessary condition to recover the dictionary, as sparse coding guarantees may not be met when the dimension $m$ is too small. We now present two examples of inverse problems with multiple operators to illustrate what happens in practice.

**Compressed sensing (CS).** When all $A_i$ are random Gaussian matrices, $(A_1^T, \cdots, A_{N_m}^T)$ is also a random Gaussian matrix of dimension $n \times N_m m$. Therefore, it is of rank $n$ with probability 1 if $N_m \geq \lfloor \frac{n}{m} \rfloor + 1$. Fig. 2 illustrates that it is indeed a necessary condition to recover $D$, but it is not sufficient when $m$ is too small, because sparse coding becomes inefficient. Multiview compressive dictionary learning has also been studied in Anaraki & Hughes (2013); Pourkamali-Anaraki et al. (2015); Chang et al. (2019).

**Inpainting.** All $A_i$ are binary masks with coefficients following Bernoulli distributions of parameters $p_1, \cdots, p_n$, i.e. $A_i = \text{diag}(a_i^1, \cdots, a_i^n)$ where each $a_i^j$ is equal to 1 with probability $p_j$. The rank of $(A_1^T, \cdots, A_{N_m}^T)$ is equal to $n$ if for each coordinate $j$ there exists an index $i$ such that $a_i^j = 1$.

This happens with probability $\prod_j (1 - (1 - p_j)^{N_m})$. Fig. A in appendix shows that similar to CS, this is a necessary but insufficient condition to recover a proper dictionary. Even when the number of samples compensates for missing values, the sparsity of the data plays a great role in the ability of the algorithm to recover the proper dictionary after heavy dimension reduction. To illustrate what happens on real data, we consider the example of image inpainting. Let $A \in \{0,1\}^{h \times w}$ be a binary mask used to observe an image $X \in [0,1]^{h \times w}$ and $Y = A \odot X$ be the observed image. While the operator is unique when we consider the whole image at once, learning a dictionary from patches of size $n$ from the image is equivalent to learning a dictionary with multiple operators in Eq. (2). Denoting $A_{ij} = \mathrm{diag}(A_{ni:n(i+1),nj:n(j+1)})$ and $Y_{ij} = \mathrm{vect}(Y_{ni:n(i+1),nj:n(j+1)})$ the $i,j$-patch, patch-based dictionary learning solves

$$\min_{Z_{ij}, D \in \mathcal{C}} \sum_{i,j} \frac{1}{2} \|A_{ij} D Z_{ij} - Y_{ij}\|_2^2 + \lambda \|Z_{ij}\|_1 \tag{5}$$

The dictionary should be recovered if the image is large enough and if there are not too many masked pixels. In Fig. 2, we show the PSNR (Peak Signal to Noise ratio) and the recovery score depending on the proportion of missing values in an image of resolution $128 \times 128$, from which we extract patches of size $10 \times 10$. To take into account that some atoms might not be as relevant as others, the score is re-weighted by the sum of corresponding activations in the sparse codes $Z$ – see Appendix A.3 for details. The recovery score drops when the proportion of missing values is larger than 50%. Otherwise, the image is successfully recovered even when the dictionary is learned from the degraded observation. This is why dictionary learning led to good results in unsupervised inpainting in the literature (Szabó et al., 2011; Studer & Baraniuk, 2012; Naumova & Schnass, 2017).

We have demonstrated that dictionary learning won't operate in the null space of the measurement matrix. However, the usage of multiple operators allows for mitigating that issue, the whole signal space being seen through different matrices $A_i$. Our experiments with synthetic and real data also show that this is only a necessary condition to learn a good dictionary. In some cases, the sparse codes cannot be recovered as the information is too degraded. Reducing the dimension of the observations could then be a hard limit to dictionary learning, and theoretical results on the convergence of classical optimization methods such as Alternating Minimization would be of great interest to ensure the identifiability of the dictionary with multiple operators. In the following, we show that well chosen weak prior knowledge can lift the problem and allow to recover the information from the kernel space of a single operator through the example of convolutions in imaging.

## 3 WEAK PRIOR KNOWLEDGE THROUGH CONVOLUTIONS

The usage of convolutions in Deep Learning (LeCun et al., 1998) has encountered tremendous success in a broad range of tasks from image classification to reconstruction. Convolutions and convolutional neural networks are efficient to analyze translation invariant data while reducing the number of parameters to be learned. In this section, we provide elements to understand the efficiency and the limitations of convolutions used as weak prior knowledge for unsupervised image reconstruction through the study of three methods based on prior learning: Convolutional Dictionary Learning (Grosse et al., 2007), Plug and Play (Chan et al., 2016) and Deep Image Prior (Ulyanov et al., 2018). All computations have been performed on a GPU NVIDIA Tesla V100-DGXS 32GB using `PyTorch` (Paszke et al., 2019).[1]

Convolutional dictionary learning (CDL) consists in learning kernels of relatively small dimensions from a signal $Y$. Lasso-based CDL solves a problem of the form

$$\min_{z_k, d_k \in \mathcal{C}} \frac{1}{2} \left\| A \sum_k d_k * z_k - Y \right\|_2^2 + \lambda \sum_k \|z_k\|_1 \quad . \tag{6}$$

Deep Image Prior (DIP) takes advantage of CNN architectures to project the observed image into a well-suited range space by drawing a random code vector $z$ in the latent space and optimizing the parameters of the network $f$ as follows

$$\min_\theta \|Y - A f_\theta(z)\|_2^2 \quad . \tag{7}$$

---

[1]Code is available in the supplementary materials.

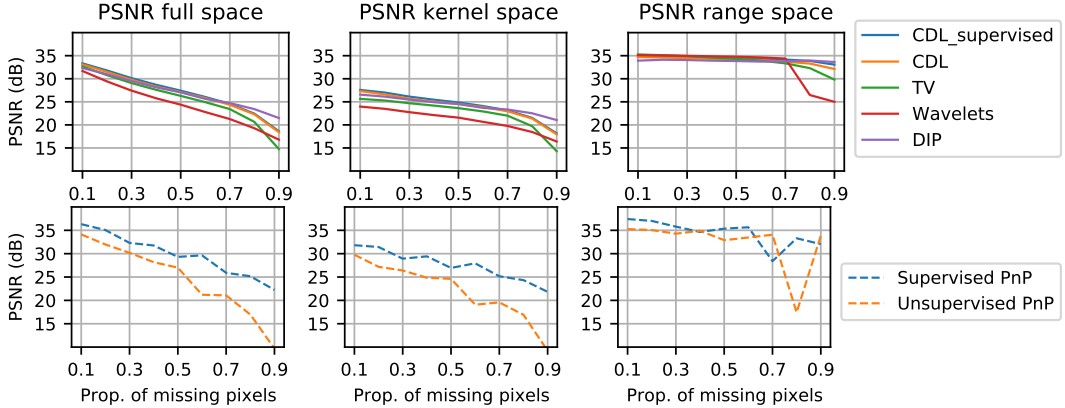

Figure 3: PSNR depending on the proportion of available pixels in the full space, kernel space, and range space of the masking operator for reconstruction methods based on CDL, DIP, TV, and wavelets for the $256 \times 256$ grey-level image in Fig. 4 (**top**), and for PnP based reconstruction on $160 \times 160$ grey-level images (**bottom**) with a SNR of 20db. Unsupervised prior learning methods work better than hand-crafted methods even with a lot of lacking information and can recover missing information in the kernel space. Moreover, they perform close to supervised methods (CDL supervised and PnP supervised) when the ratio of missing pixels is not too high.

Plug and Play (PnP) is an iterative algorithm inspired from proximal gradient descent, which recovers images from an observation $Y$ with steps of the form

$$X_{n+1} = f_\theta \left( X_n - \tau A^*(A X_n - Y) \right) \quad \forall n \geq 1 \; , \tag{8}$$

where $X_0 = 0$, $\tau$ is a step size and $f_\theta$ is an image denoiser. CDL and DIP can be applied to a single observation without training on a data set, the prior being learned directly on one piece of degraded data without needing other information. In contrast, PnP usually resort to a deep denoiser generally pre-trained on a clean database. As we focus on the unsupervised setting, we adapt PnP by training the denoiser on degraded data instead. In this case, we consider that we have access to a dataset $(Y_i)_{1 \leq i \leq N}$ where each $Y_i = A X_i + \epsilon_i$ is an observation of an original image $X_i$ degraded by the same operator $A$ and a gaussian noise $\epsilon_i$. We artificially generate noisy images $(Y_i')_{1 \leq i \leq N}$ from our data-set of observations $Y_i' = Y_i + \epsilon_i'$, and we train a DnCNN (Zhang et al., 2017) to recover $Y_i$ from $Y_i'$ in the range space of $A$ by minimizing

$$\min_\theta \frac{1}{N} \sum_i \|A(f_\theta(Y_i') - Y_i)\|_2^2 \; . \tag{9}$$

The idea is to check in which case the architecture can compensate for the lack of information in the kernel of $A$ by learning from the information in the range space of $A$. To point out the limits of these prior learning algorithms, we will compare them to two reconstruction methods based on Total Variation (TV) (Chambolle et al., 2010) and sparse wavelets (Mallat, 2008).

The purpose is to highlight the hard limits of unsupervised methods in various contexts. Therefore, we evaluate the performance reached by each algorithm over oracle hyper-parameters, namely hyper-parameters leading to the best performances. While evaluating hyper-parameter sensitivity is necessary when comparing different methods, it is orthogonal to our study, which considers the difference between supervised and unsupervised training of similar methods.

### 3.1 WHY CONVOLUTIONS ARE LIKELY TO WORK ON TASKS LIKE INPAINTING

Works on prior learning in unsupervised inverse problems often evaluate the performance of the methods they propose on an inpainting task (Ulyanov et al., 2018; Chen et al., 2021) and achieve very good performance compared to supervised learning techniques. Here, we provide elements to understand why this problem, in particular, is feasible with the help of convolutional dictionaries or neural networks without access to ground truth data.

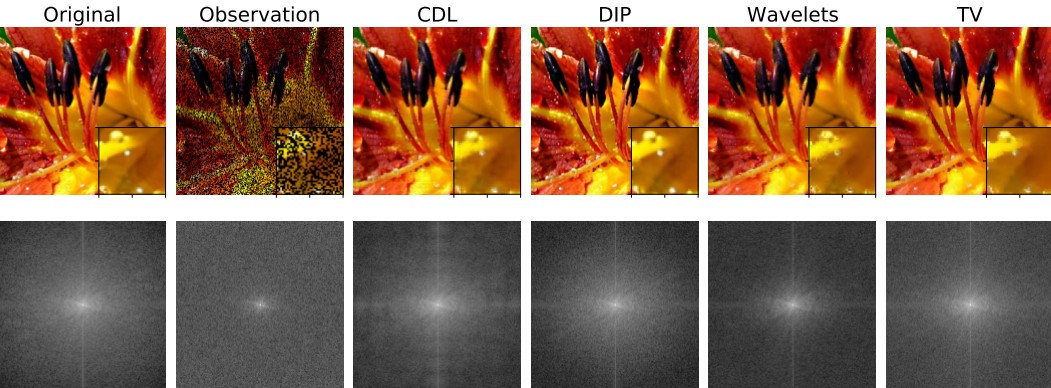

Figure 4: Reconstruction and PSD of a $256 \times 256$ RGB image with 50% missing pixels in a noiseless scenario. PSNR: CDL 34.8dB, DIP 34.1dB, Wavelets 34.7dB, TV 34.3dB. The PSD reveals the presence of ringing artifacts in the reconstruction by CDL. Otherwise, unsupervised algorithms recover the whole spectrum of the original image and do as well as hand-crafted methods.

**Learning convolutional dictionaries from incomplete data.** To understand what happens in inpainting, let's consider a simple one-dimensional signal example. Let $X_t$ be a wide sense stationary (WSS) random process, and let $A_t$ be an i.i.d Bernoulli process of mean $\rho$. The observed signal $Y_t = A_t X_t$ is also a WSS random process and its auto-correlation function $R_Y(\tau)$ is

$$R_Y(\tau) = \mathbb{E}[A_t X_t A_{t+\tau} X_{t+\tau}] = R_X(\tau)\mathbb{E}[A_t A_{t+\tau}] \tag{10}$$

$$= \rho^2 R_X(\tau)\mathbb{1}_{\tau \neq 0} + \rho R_X(\tau)\mathbb{1}_{\tau=0} \; . \tag{11}$$

Then, the Wiener-Khintchine theorem assures that the power spectral density of $X$ and $Y$ are proportional. This shows that with sufficient samples in the signal, the masking process won't affect the spectrum of the original signal $X$, and translation invariant priors can take advantage of the information from all frequencies.

We illustrate the practical implication of this observation on the ability of CDL to recover 10 digits from an image, depending on the size of the image and the rate of available pixels $\rho$ in Fig. 5. As expected, the performance increases with the size when $\rho$ is not too low. It is essential to note that having access to all frequencies is only a necessary condition to learn a good dictionary, as sparse coding assumptions are not met when there are too many missing values. Of course, these results do not stand for non-stationary signals.

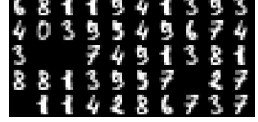

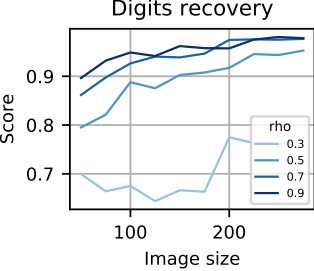

Figure 5: The recovery score of CDL depends on the image size and the rate of available pixels $\rho$. Increasing the size improves the quality at high enough rates.

**Unsupervised reconstruction.** Similar effects can be observed for reconstruction. Natural images are stable enough to allow convolution-based algorithms to learn from all frequencies that are present in the signal. Fig. 3 presents an example where a single natural image is degraded by a random binary mask and gaussian noise and reports the PSNR of the reconstruction in the mask kernel space and range space for CDL, DIP, and methods based on TV and sparse wavelets for different rates of missing pixels with a SNR of 20dB. Other experiments with different values of SNR are available in appendix in Fig. C. Supervised means the algorithm learns a prior on the clean signal and uses it for reconstruction after degradation, whereas unsupervised means that the prior is learned directly on the observation. The experiments highlight that unsupervised methods work as well as supervised CDL and are better than hand-crafted priors in the kernel of $A$ when the noise level is not too high (SNR $\geq 20$). They succeed in learning in the range space of $A$ and generalizing in the kernel space. The PSNR drops in favor of TV and wavelets when the noise increases, as it becomes more challenging to learn the structure of the signal. Fig. 4 provides a visual example in the noiseless case. Unsupervised algorithms successfully recover the original image after degradation by a binary mask with 50% pixels missing. The PSD shows that

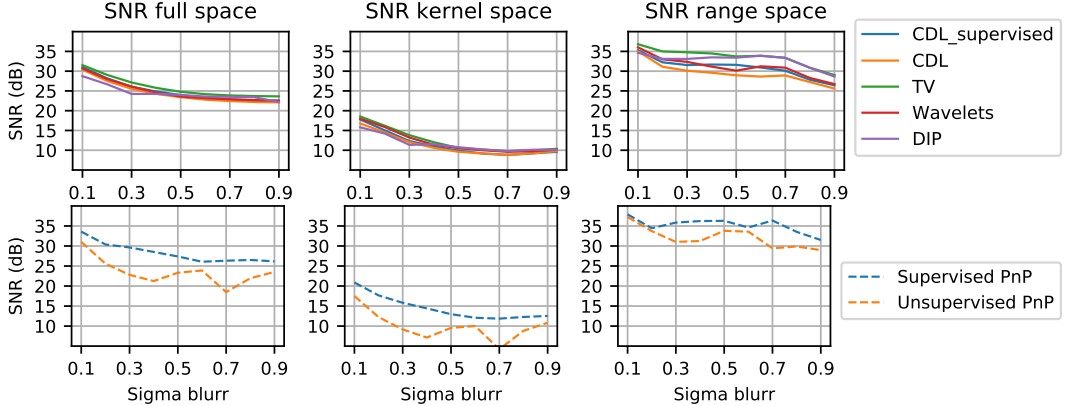

Figure 6: SNR depending on the size of the blur in the full space, kernel space, and range space of the blur operator for reconstruction methods based on CDL, DIP, TV, and wavelets for th 256 × 256 grey-level image in Fig. 7 (**top**), and for PnP based reconstruction on 160 × 160 grey-level images (**bottom**) with a SNR of 20db. This time, unsupervised prior learning methods fail to recover information in the kernel space. More surprisingly, supervised PnP and CDL also struggle in the kernel space.

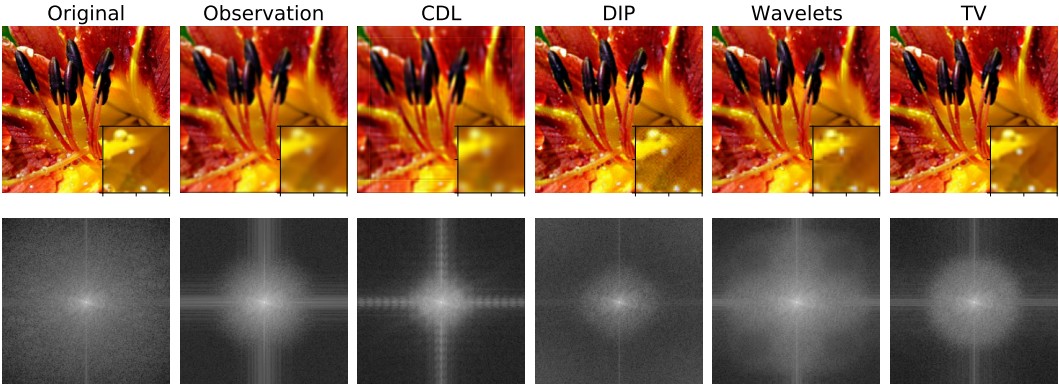

Figure 7: In a noiseless scenario, reconstruction and PSD of a 256 × 256 RGB image blurred by a gaussian kernel. PSNR: CDL 31.9dB, DIP 31.7dB, Wavelets 34.6dB, TV 32.7dB. The PSD clearly shows that nothing is learned in high frequencies compared to what was obtained in inpainting.

low and high frequencies are retrieved, despite ringing artifacts in the case of CDL. However, CDL and DIP are sensitive to noise and fail to recover relevant frequencies from the observations in noisy scenarios – see Fig. B in the appendix.

In the case of Plug-and-Play, we train a DnCNN to recover noisy images from the dataset Imagenette[2] and plug it into an iterative reconstruction algorithm. In the unsupervised case, the denoiser learns how to recover degraded images, as explained above. The results are shown in Fig. 3 for SNR=20dB and in Fig. D in appendix for more values of SNR. When the noise is low, i.e., SNR $\geq$ 20, and when the rate of missing pixels stays below 50%, unsupervised and supervised PnP leads to similar performance levels in terms of PSNR. As for the single image example, unsupervised PnP can generalize what is learned in the range space of $A$ to the kernel space and performs closely to its supervised counterpart as long as the rate of masked pixels is not too large.

## 3.2 THE PITFALL OF CONVOLUTIONS IN DEBLURRING

Convolutions work well when all frequencies are preserved, as shown for inpainting. However, several inverse problems involve recovering a signal with missing frequencies. In super-resolution,

[2]The data are available at https://github.com/fastai/imagenette

all odd frequencies lack in the signal. In deblurring, the signal is observed after degradation by a low-pass filter. Unsupervised prior learning becomes troublesome in these cases, as mentioned in (Tachella et al., 2022). We will focus on the example of deblurring in the following.

The CDL problem can be re-written in terms of Fourier transforms with the Parseval equality

$$\min_{z_k, d_k} \frac{1}{2} \left\| \hat{A} \sum_k \hat{d}_k \hat{z}_k - \hat{Y} \right\|_2^2 + \lambda \sum_k \|z_k\|_1 \quad . \quad (12)$$

As the spectrum $\hat{A}$ is low-pass, nothing is observed in high frequencies. Thus, optimal dictionaries contain atoms $(d_k)_k$ with high frequencies set to $0$, for the same reason as pointed out in Proposition 2.1. This is illustrated in Fig. G in appendix.

Fig. 7 displays the reconstructions and PSD of a blurred image for various methods and Fig. 6 their performances for various blur sizes in the kernel and range spaces. These results show that neither CDL nor DIP can recover information outside the span of the blur, i.e., in high frequencies. While CDL puts all high frequencies to $0$, DIP adds noise. The same phenomenon appears with PnP: there is a performance gap between supervised and unsupervised learning in the kernel, and generalization from the range space to the kernel space is impossible.

**Is self-supervised learning adapted to deblurring ?** We have demonstrated why unsupervised prior learning based on convolutions won't work on deblurring. The figures also show that learning a dictionary or a denoiser on clean signals does not seem to lead to convincing performance: supervised CDL does not outperform hand-crafted priors, and supervised PnP hardly does better in the kernel space. The key to success is finding a prior that can predict missing high frequencies from observed low frequencies. Indeed, the average PSD output by the DnCNN on Fig. 8 shows that the denoiser spontaneously learns to add high frequencies when trained on images that have not been blurred, which is not the case otherwise. Then we look at the discrepancy of frequencies in atoms learned on a clean image with CDL, defined as

$$\sum_{i,j} \mathrm{PSD}(\omega_i)\mathrm{PSD}(\omega_j)|\log(\omega_i + 1) - \log(\omega_j + 1)| \quad . \quad (13)$$

Atoms with high discrepancies contain a broad range of frequencies, while the others only focus on specific bandwidths.

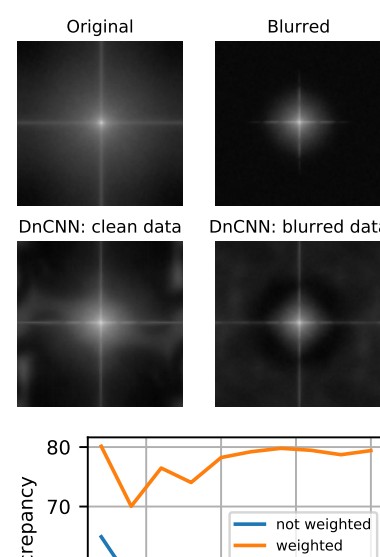

Figure 8: (**top**) Average PSD of 10 iterations of DnCNN trained on clean and corrupted data. When trained in a supervised setting, the denoiser spontaneously adds high frequencies to the image. (**bottom**) Discrepancy of frequencies in atoms from supervised convolutional dictionaries. When their number increases, the atoms do not necessarily mix frequencies in general (not weighted), but the ones used by sparse coding (weighted) integrate high and low frequencies.

We show the average and the weighted average to take into account the usage that is made of these atoms. It is interesting to note that the algorithm prefers atoms with large bandwidths.

These experiments highlight that learning priors on pretext tasks like denoising in a supervised setting somehow captures the spectral structure. However, supervised CDL and PnP do not perform as well as expected on deblurring. This hints that it may be necessary to add specific regularizations or non-linearities like Sharpen filters (Habeeb et al., 2018) to these models to ensure that they learn how to link low and high frequencies appropriately.

## 4 CONCLUSION

Prior learning for unsupervised inverse problems is only feasible with multiple operators or appropriate constraints in the model. When the operator is too ill-conditioned, which is typically the case in deblurring, the prior knowledge should compensate for the lack of information. In particular, exclusively relying on convolutions is not enough, even in self-supervised settings with access to clean data.

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

## A  FULL DESCRIPTION OF THE EXPERIMENTS

This section provides complementary information on the experiments presented in the paper.

**Optimization for dictionary learning.**  The Lasso is always solved with FISTA (Beck & Teboulle, 2009), then the minimization is done by gradient descent over the dictionary with the help of a line search.

### A.1  PARTIAL RECOVERY IN DICTIONARY LEARNING WITH A SINGLE OPERATOR - FIG. 1

We generate $10\,000$ samples from a random Gaussian dictionary $D$ of dimension $100 \times 100$ and Bernoulli Gaussian sparse codes $Z$ with several levels of sparsity, i.e. ratio of non zero elements in average. Then we degrade the signals with a random Gaussian compressed sensing operator of dimension $m \times 100$ for several values of $m$, i.e. $Y = ADZ$. We compare the maximal recovery scores over $\lambda$ for each level of sparsity with the perfect score that we can achieve in the range space of a single operator. We repeat this experiment 5 times.

### A.2  RECOVERY IN DICTIONARY LEARNING WITH MULTIPLE OPERATORS - FIG. 2

We generate $N = 10\,000$ samples from a random Gaussian dictionary $D$ of dimension $100 \times 100$ and Bernoulli Gaussian sparse codes $Z$ with a ratio of non zero elements of $0.1$. Then we degrade the signals with a given number $N_m$ of random Gaussian compressed sensing operators of dimension $m \times 100$, i.e. samples $[i\lfloor\frac{N}{N_m}\rfloor, (i+1)\lfloor\frac{N}{N_m}\rfloor[$ are seen through operator $A_i$. We compare the maximal recovery scores over $\lambda$ for each $N_m$ with the perfect score that we can achieve with a single operator. We repeat this experiment 5 times.

### A.3  RECOVERY IN DICTIONARY LEARNING ON PATCHES FROM A NATURAL IMAGE IN INPAINTING - FIG. 2

We mask a grey-level $256 \times 256$ image with a binary mask containing a given level of missing pixels. Then we split this image into $10 \times 10$ patches and we learn a dictionary of size $100$ on these patches, with $\lambda = 0.1$. We compute the recovery score by comparing dictionaries obtained with and without the mask, and we weight the costs by the average of sparse activations $Z$ in absolute value for a given atom

$$W = (\sum_{i=1}^{N} |Z_{1,i}|, \cdots, \sum_{i=1}^{N} |Z_{L,i}|)^T / \sum_{i,j} |Z_{i,j}| \tag{14}$$

$$C = D_0^T (D^T \odot W)^T \ . \tag{15}$$

This score better reflects the usefulness of the atoms, and allows to take into account the fact that dictionaries learned on natural signals may contain irrelevant atoms that are almost never used. We repeat this experiment 10 times.

### A.4 PSNR RECONTRUCTION IN INPAINTING AND DEBLURRING - FIG. 3 AND FIG. 6

**Single image (top).** We mask a grey-level $256 \times 256$ image with a binary mask containing a given level of missing pixels.

- CDL. We learn a convolutional dictionary on the observed image (unsupervised) or clean image (supervised) with atoms of size 10, and we take the maximal PSNR over several values of $\lambda$.

- DIP. We leverage the architecture based on SkipNet proposed in Ulyanov et al. (2018), and we train the network as in the DIP paper with Adam (Kingma & Ba, 2014), a learning rate of 0.01, an input noise of standard deviation $\sigma = 0.1$, and a regularization noise of standard deviation $\sigma = 0.03$. The result is displayed after 1000 epochs. This choice of hyper-parameters led to good results in practice.

**Plug and Play (bottom).** The dataset is composed of 1000 natural images from the dataset Imagenette available at https://github.com/fastai/imagenette in grey-level and cropped to size $160 \times 160$. We mask the images with the same operator $A$, and we learn to denoise degraded images $Y_i'$ by minimizing the loss

$$\frac{1}{N} \sum_{i=1}^{N} \|A(f_\theta(Y_i') - Y_i)\|_2^2 \tag{16}$$

where $Y_i' = Y_i + \epsilon_i$ with $\epsilon_i \sim \mathcal{N}(0, \sigma^2)$. The network is a DnCNN (Zhang et al., 2017) available at https://github.com/cszn/DnCNN.git We train it wih Adam, learning rate 0.001, batch size 32, maximal number of epochs 50. Then the network $f_\theta$ is integrated in a reconstruction algorithm

$$X_{n+1} = f_\theta(X_n - \tau A(X_n - Y)) \tag{17}$$

for 100 iterations, with $\tau = 1$, which led to good results in practice. We display the maximal average PSNR on the test set composed of 50 images from Imagenette degraded in the same way, over $\sigma$ and epochs.

**SNR for deblurring.** We rely on the Signal to Noise ratio (SNR) to measure the performance in the case of deblurring. We define the range space of the blur as the part of its spectrum within $2\sigma$. 95% of the power is contained in the range space, and 5% in the kernel space, hence the interest in switching to the SNR instead of the PSNR.

### A.5 DIGITS RECOVERY WITH DICTIONARY LEARNING - FIG. 5

We generate an image composed of 10 digits from the dataset `digits` of `scikit-learn` (Pedregosa et al., 2012). We then learn a convolutional dictionary with 30 atoms of size $10 \times 10$, and look at the correlation between each atom and each digit (we keep the best for each digit) to compute the average recovery on all digits. The algorithm is run with different values of hyper-parameters and we keep the best recovery score (oracle predictor). The experiment is repeated for several sizes of images, from $50 \times 50$ to $300 \times 300$ and for several rates of missing pixels, with gaussian noise of standard deviation $\sigma = 0.1$.

### A.6 ILLUSTRATION OF INPAINTING AND DEBLURRING - FIG. 4, FIG. 7 AND FIG. B

In the case of inpainting, the image is degraded by a binary mask with a rate of missing pixels of 50% without noise in Fig. 4 and with SNR=6dB in Fig. B In the case of deblurring, the image is degraded by a gaussian blurr of size $10 \times 10$ with $\sigma = 0.3$ without noise. The algorithms that can be used on a single image are run on each channel as described in Fig. 3 and we display the reconstruction plus the power spectrum density.

## A.7 SELF-SUPERVISED PRIOR LEARNING - FIG. 8

**Plug and Play (Top).** We train a DnCNN as explained for Fig. 6 on both clean (supervised) and blurred (unsupervised) images (1000 samples), and we look at the average PSD of images output by 10 fixed point iterations of the network when fed with the test set (50 samples), i.e.

$$X_{n+1} = f_\theta(X_n) \quad 1 \le n \le 10 \ , \tag{18}$$

where $X_0$ is the blurred observation. The goal is to check if the network is able to add missing high frequencies to an image after being trained on a pretext task like denoising in the context of Plug and Play, where gradient descent steps alternate with forward passes in the denoiser.

**CDL (Bottom).** We learn a convolutional dictionary with atoms of size $20 \times 20$ on a clean grey-level image of size $256 \times 256$ and measure the discrepancy of frequencies in the atoms depending on the number of atoms in the dictionary. In this example, the level of sparsity is high (we set $\lambda = 1$). The discrepancy is defined as

$$\sum_{i,j} \text{PSD}(\omega_i) \text{PSD}(\omega_j) |\log(\omega_i + 1) - \log(\omega_j + 1)| \ . \tag{19}$$

We repeat the experiment 20 times.

## B EXTRA FIGURES AND EXPERIMENTAL RESULTS

### B.1 DICTIONARY RECOVERY DEPENDING ON SPARSITY, PROPORTION OF MISSING VALUES AND NUMBER OF SAMPLES - FIG. A

We generate the data from a Gaussian dictionary of size $100 \times 100$ and Bernoulli Gaussian sparse codes of sparsity $s$ (average rate of non zero coordinates). Then, we degrade the data with a binary mask of variable rates of available coordinates $p$. We learn a dictionary of size $100 \times 100$ over several values of $\lambda$. We show the ability of the algorithm to recover the dictionary, depending on $p$, the number of training samples and the level of sparsity in the data. Dictionary recovery is defined as obtaining a recovery score of at least 0.95. We display the results from three different perspectives:

- Left. Minimal number of samples necessary to recover the dictionary depending on sparsity and rate of available coordinates in the data. A number of samples larger than $10^4$ means no recovery possible.
- Center. Maximal level of sparsity $s$ (maximal proportion of non zero coordinates) to recover the dictionary depending on the number of samples and the rate of available coordinates. A level equal to 0 means no recovery possible.
- Right. Minimal rate of available coordinates for recovery depending on the number of samples and the level of sparsity. A level equal to 1 means no recovery possible.

These figures show that there is a hard limit to what can be learned depending on the proportion of missing values and sparsity, regardless the number of training samples. Having access to the whole signal space is not a sufficient condition to recover the dictionary.

### B.2 RECONSTRUCTION WITH SNR 6DB - FIG. B.

The image is degraded by a binary mask with a rate of missing pixels of 50% and with gaussian noise such that the SNR is equal to 6dB. The algorithms that can be used on a single image are run on each channel as described for Fig. 4 and we display the reconstruction plus the power spectrum density. As opposed to what we observed for SNR=20dB, the reconstruction is made harder by the presence of noise.

### B.3 PSNR FOR INPAINTING WITH VARIOUS LEVELS OF NOISE - FIG. C AND FIG. D

We repeat the experiments from Fig. 3 with various levels of SNR: noiseless, 20dB, 12dB, 6dB. When the noise is too high, unsupervised prior learning methods fail to learn a proper prior in the range space of $A$.

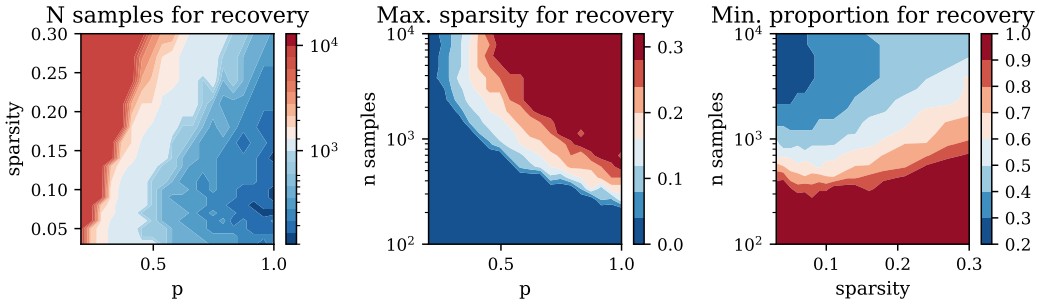

Figure A: Dictionary recovery depending on sparsity, proportion of available coordinates $p$ and number of samples. There is a hard limit to what can be learned in inpainting, which depends on $p$ and sparsity.

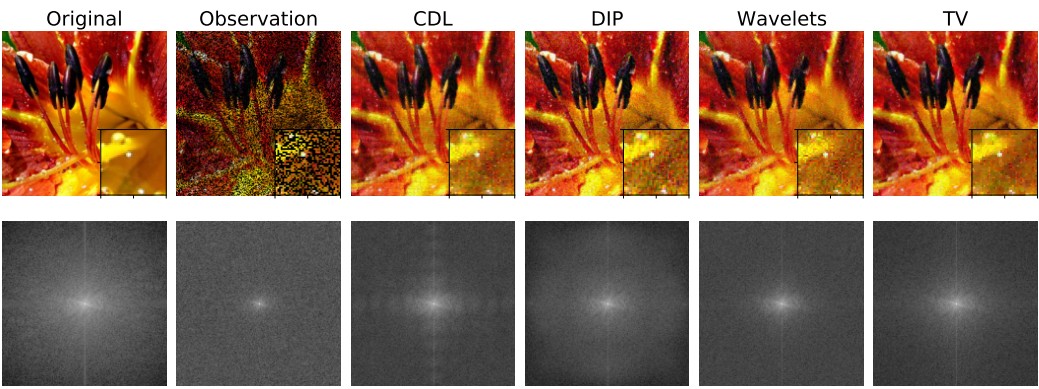

Figure B: Reconstruction in a noisy scenario with SNR 6dB.

## B.4 SNR FOR DEBLURRING WITH VARIOUS LEVELS OF NOISE - FIG. E AND FIG. F

We repeat the experiments from Fig. 6 with various levels of SNR: noiseless, 20dB, 12dB, 6dB. When the noise is too high, unsupervised prior learning methods fail to learn a proper prior in the range space of $A$.

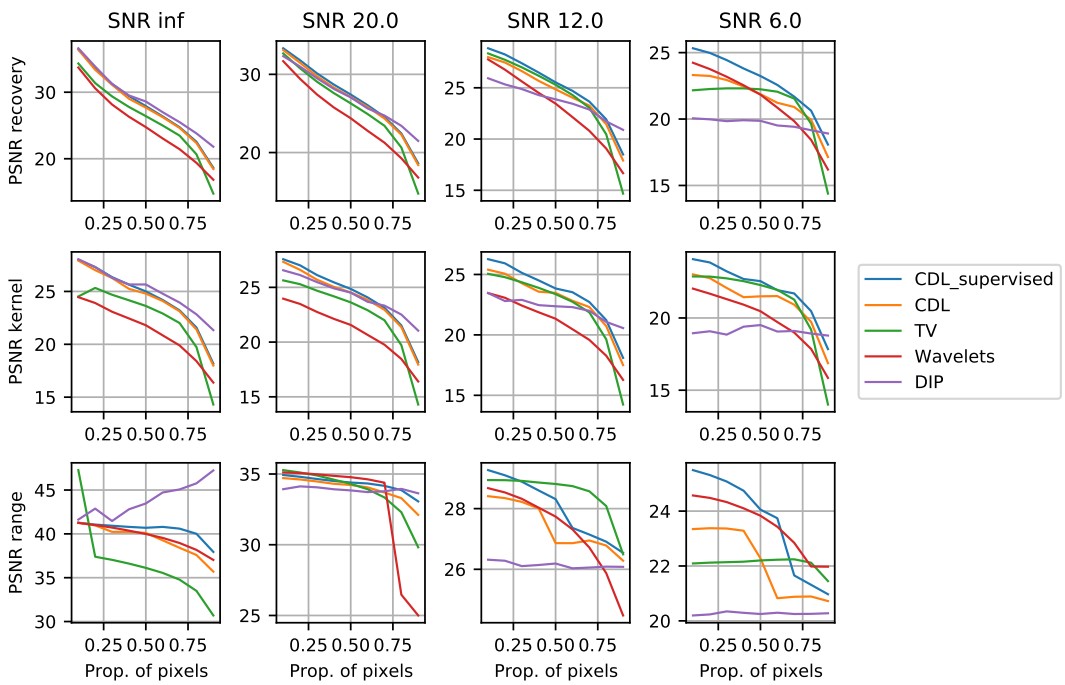

Figure C: PSNR depending on the proportion of available pixels and noise for CDL, DIP, TV and wavelets based reconstruction on a $256 \times 256$ grey-level image. Unsupervised prior learning works only when the noise is not too high.

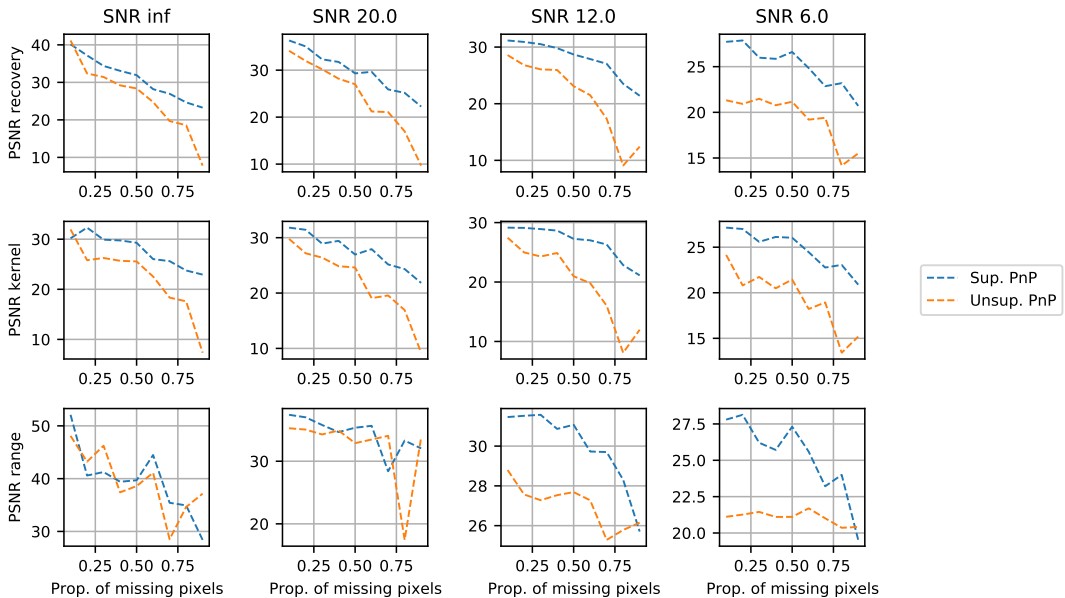

Figure D: PSNR depending on the proportion of available pixels and noise for supervised and unsupervised Plug and Play. When the noise is too high, unsupervised PnP fails to recover the image both in the kernel and the range space of $A$.

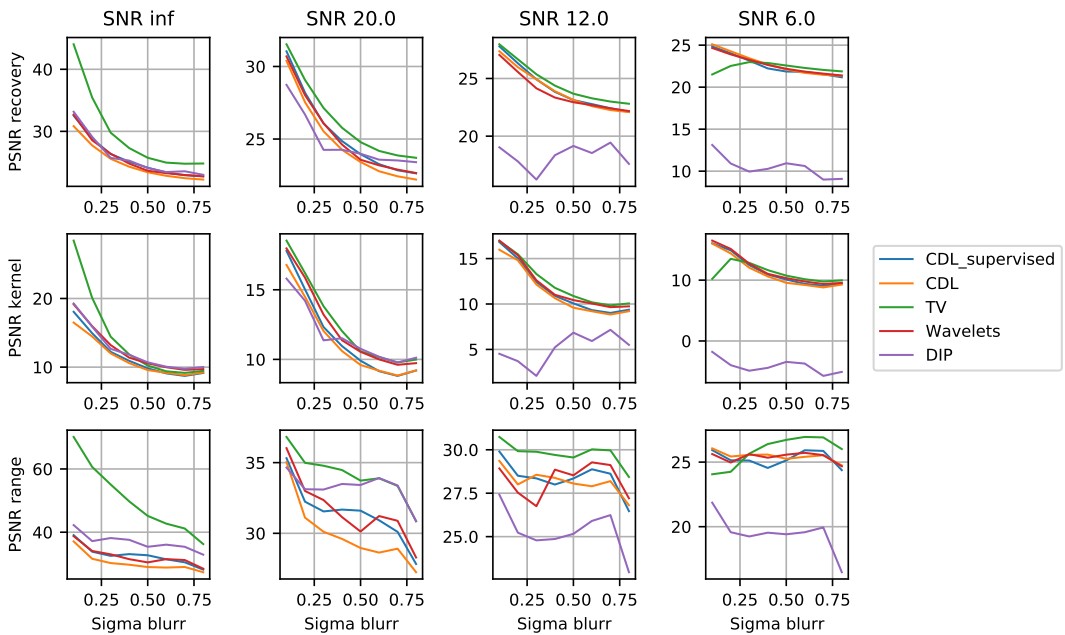

Figure E: SNR depending on the standard deviation of the blurr and noise for CDL, DIP, TV and wavelets based reconstruction on a $256 \times 256$ grey-level image. Noise makes it difficult for unsupervised prior learning to work in the range space of $A$.

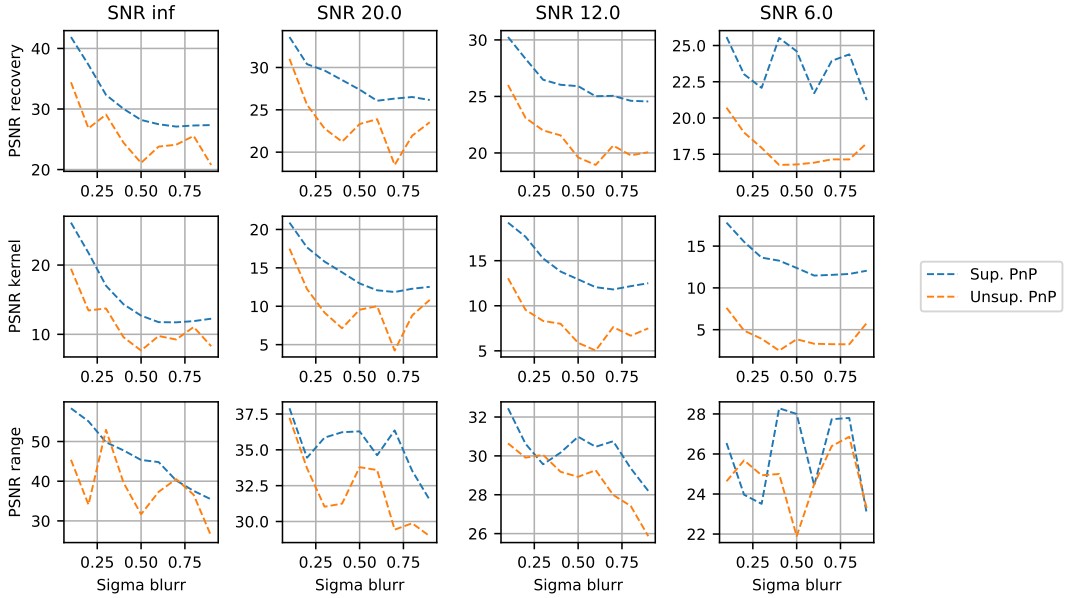

Figure F: SNR depending on the standard deviation of the blurr and noise for supervised and unsupervised Plug and Play. When the noise is too high, unsupervised PnP fails to recover the image in the range space of $A$.

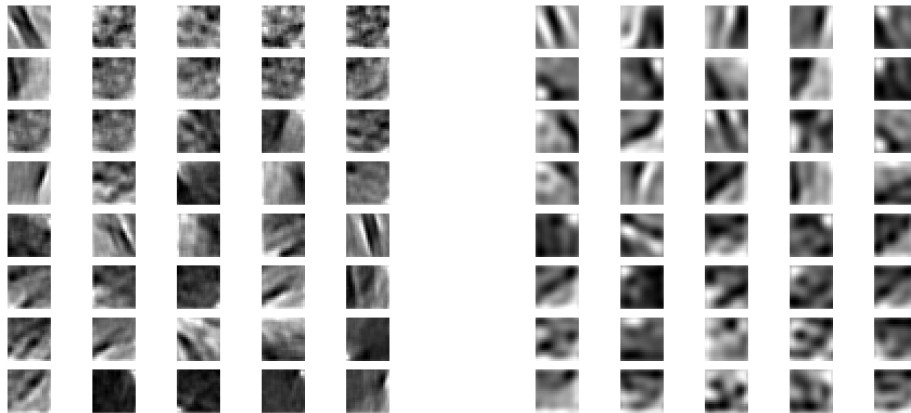

Figure G: 40 atoms of a dictionary learned with CDL ($\lambda = 0.1$, atoms of size $20 \times 20$) for inpainting (left) and deblurring (right) on the image of Figure 4. High frequencies are put to 0 in the case of deblurring, thus the atoms appear blurry.

## C Proofs

### C.1 Proof of Proposition 2.1

Let $Z_0 \in \arg\min_Z \frac{1}{2}\|AD_0 Z - Y\|_2^2 + \lambda\|Z\|_1$. Let $Z_j' = \frac{\|D_{0,j,m}\|}{\|D_{0,j}\|} Z_{0j}$. Then

$$\|AD'Z' - Y\|_2 = \|AD_0'Z_0' - Y\|_2 \tag{20}$$

$$\|Z'\|_1 \leq \|Z_0\|_1 \tag{21}$$

The result follows.

### C.2 Proof of Proposition 2.2

Let $A \in \mathbb{R}^{m \times n}, Y \in \mathbb{R}^{m \times T}$. We aim to solve

$$\min_{D \in \mathcal{C}, Z} \frac{1}{2}\|ADZ - Y\|_2^2 + \lambda\|Z\|_1 \tag{22}$$

Performing a SVD on A leads to

$$A = U\Lambda V^* \text{ s.t. } U \in \mathbb{R}^{m \times m}, V \in \mathbb{R}^{n \times n} \text{ and } UU^* = I_m, VV^* = I_n \tag{23}$$

$$\Lambda = \begin{bmatrix} \lambda_1 & 0 & \cdots & 0 & \cdots & 0 \\ 0 & \lambda_2 & \cdots & 0 & \cdots & 0 \\ \vdots & \vdots & \ddots & \vdots & \vdots & \vdots \\ 0 & 0 & \cdots & \lambda_m & \cdots & 0 \end{bmatrix} \tag{24}$$

Then,

$$\min_{D \in \mathcal{C}, Z} \frac{1}{2}\|ADZ - Y\|_2^2 + \lambda\|Z\|_1 = \min_{D \in \mathcal{C}, Z} \frac{1}{2}\|U\Lambda V^* DZ - Y\|_2^2 + \lambda\|Z\|_1 \tag{25}$$

$$= \min_{D \in \mathcal{C}, Z} \frac{1}{2}\|\Lambda V^* DZ - U^* Y\|_2^2 + \gamma\|Z\|_1 \tag{26}$$

$$= \min_{\tilde{D} \in \mathcal{C}, Z, D = V\tilde{D}, \tilde{Y} = U^* Y} \frac{1}{2}\left\|\Lambda \tilde{D} Z - \tilde{Y}\right\|_2^2 + \gamma\|Z\|_1 \tag{27}$$

Adding zeros to $\Lambda$ to make it square, and adding zeros at the end of the measurement vector $U^* Y$ to respect dimensions, the problem reduces to

$$\min_{D \in \mathcal{C}, Z} \frac{1}{2}\left\|\Lambda \tilde{D} Z - \tilde{Y}\right\|_2^2 + \lambda\|Z\|_1 \text{ s.t. } \Lambda = \text{diag}(\lambda_1, \cdots, \lambda_m, 0, \cdots, 0), \tilde{Y} = \begin{pmatrix} U^* Y \\ 0_{n-m} \end{pmatrix} \quad . \tag{28}$$

Then, Proposition 2.1 applies and an optimal dictionary is contained in $\ker(A)^\perp$.

