# OpenReview forum: "Where prior learning can and can't work in unsupervised inverse problems"
_ICLR.cc/2023/Conference — Submitted to ICLR 2023_

### Official Review · Reviewer_aFy7 · 2022-10-24

**Confidence:** 2
**Correctness:** 4
**Technical Novelty And Significance:** 3
**Empirical Novelty And Significance:** 3
**Recommendation:** 6

**Clarity, Quality, Novelty And Reproducibility:**

The presentation is clear. The questions are well posed and clearly answered, providing clear claims and discussions for the readers.

**Strength And Weaknesses:**

Weakness: this paper is roughly an extension of the work conducted by Tachella et al. (2022)

Strengths:
The paper is well written, clearly presenting the claims and discussions. The paper presents many interesting results, within different tasks such as inpainting and deblurring. See other comments for details

**Summary Of The Paper:**

This paper studies where (weak) prior learning can and can’t work in unsupervised inverse problems, by considering dictionary learning in a conventional model as well as convolutional dictionary learning. Inpainting and deblurring asks are addressed in this work.


**Summary Of The Review:**

Essentially, this work extends the study conducted by Tachella et al. (2022) in order to provide a practical understanding of when one can and can’t learn a prior for unsupervised inverse problems. In the submitted paper, the authors focus on the case where the prior is not constrained, with the analysis of dictionary learning, using a single measurement operator first and then extending these results to multiple operators. Finally, they address prior knowledge within convolutional dictionary learning, and demonstrated several claims, such as why convolutions are likely to work on tasks like inpainting, including learning convolutional dictionaries from incomplete data as well as the unsupervised reconstruction. The deblurring task was also considered in the paper.
The claims are clearly presented and going to the main claims, such as the prior knowledge should compensate for the lack of information in deblurring tasks.

---

> ### Author Response · Authors · 2022-11-15
> **Answer to reviewer aFy7**
>
> Thank you for this positive feedback which reflects well the main contributions. As stated in the general answer, we also complement the results from Tachella et al. (2022) with many experiments and illustrations that will reinforce this important message in the inverse problem community.

---

### Official Review · Reviewer_ZBdm · 2022-10-25

**Confidence:** 4
**Correctness:** 4
**Technical Novelty And Significance:** 2
**Empirical Novelty And Significance:** 3
**Recommendation:** 6

**Clarity, Quality, Novelty And Reproducibility:**

Clear and written well. It has nice visualization and explanation of their method. Some of the discussed properties/characterizations are known in the literature, and not clear what is new (see my comment about novelty and a need for discussion). They provided detailed info and code on the results, hence it seems reproducible.

**Strength And Weaknesses:**

The paper is well organized. I enjoyed reading the paper. It provides very good analysis, nice explanations to give intuitions (e.g., paragraph after (4)), and clear experiments on when and where supervised/unsupervised inverse problems may fail.

- The paper points out interesting insights on why dictionary learning with one operator may fail. It would be nice to indeed cite works that use several (instead of one) random measurement matrices for compressive dictionary learning (here are three examples [1,2,3]) when you introduce compressive dictionary learning (1).

- My main concern is that it is not clear which result is known from prior work and which is theirs. Hence, please discuss the novelty of the work and how it is different from prior work. I find this paper more of a review paper and summarizing what is known in the literature. Please make a clear discussion on the new finding of this paper compared to prior work. For example, some of the findings of this paper are based on Tachella et al. (2022).

- The paper considers only dictionary learning and linear inverse problem. How generalizable is this to other generative models and for example non-linear inverse problems?

minor comments:
- Before (4), "if this is not feasible, then (2) is not feasible". Please explain why?
- It would be nice to visualize some of the dictionaries learned in this work.
- Although very well-known, please cite lasso before (1) [4].
- Move figures close to their explanation.
- please use the same y-axis in sub-figures (e.g., page 16).

[1] Anaraki, Farhad Pourkamali, and Shannon M. Hughes. "Compressive k-svd." 2013 IEEE International Conference on Acoustics, Speech and Signal Processing. IEEE, 2013.

[2] Pourkamali-Anaraki, Farhad, Stephen Becker, and Shannon M. Hughes. "Efficient dictionary learning via very sparse random projections." 2015 International Conference on Sampling Theory and Applications (SampTA). IEEE, 2015.

[3] Chang, Thomas, Bahareh Tolooshams, and Demba Ba. "Randnet: Deep learning with compressed measurements of images." 2019 IEEE 29th International Workshop on Machine Learning for Signal Processing (MLSP). IEEE, 2019.

[4] Tibshirani, Robert. "Regression shrinkage and selection via the lasso." Journal of the Royal Statistical Society: Series B (Methodological) 58.1 (1996): 267-288.


**Summary Of The Paper:**

The paper focuses on unsupervised inverse problems, and studies under what conditions recovery is possible. Their contribution involves a series of characterization and analyses of the problem, explained below.

- They focus on the compressive dictionary learning problem and provide the conditions under which dictionary recovery is possible given one or multiple compression operators (measurement matrix).
- They focus on convolutional structure as a weak prior, and show how unsupervised CDL is able to perform similarly to supervised CDL when compression is not high.
- They show that for deblurring, where high-frequency information is completely lost, supervised CDL, unsupervised CDL, and all the hand-crafted features fail to recover.
- Given all these analyses and characterization of unsupervised inverse problems, they phrase the ultimate goal as "the key to success is finding a prior that can predict missing high frequencies from observed low frequencies." They show that the structure of DnCNN is indeed able to do that by capturing spectral information.


**Summary Of The Review:**

I enjoyed reading the paper. It is an interesting work on analyzing why unsupervised inverse problems may fail, how to bake weak priors to alleviate that, how supervised inverse problems may fail, and how to apply methods to capture spectral information to recover information that seems to be lost from the perspective of classical optimization. However, it is not clear if these results are already known in prior works. A discussion is needed here. Given that, I rate this paper as marginal acceptance. I may change my rating depending on the requested discussion.

---

> ### Author Response · Authors · 2022-11-15
> **Answer to ZBdm**
>
> **Related work -** Thank you for pointing out these references. Using several random measurement matrices is indeed directly related to this work. We will make sure to cite [1,2,3] as well as the lasso [4].
>
> **Contributions -** Please see our general response on the novelty and implications of our work. Summarizing, explaining and illustrating what is known in the literature is indeed one of our contributions. We will make our contributions clearer in the revised manuscript, posted later this week, as well as clarify our discussion in the limited space that we have.
>
> **Generalization to non-linear inverse problems -** The focus was indeed on linear inverse problems, and we have not tried to generalize it to other models yet. While a generalization of our work to a broader class of inverse problems would be of interest, it would first require extending the notion of kernel space to non-linear models, and it is not trivial to extend our analysis in this case, as it is based on linear algebra properties.
>
> **Precisions on the gradient study in dictionary learning -** Let $(g_i)_{1 \le i \le N}$ be gradient estimates obtained without $A_i$, i.e. $g_i = (DZ_i - X_i)Z_i^T$ where $X_i$ is the data point before degradation by $A_i$. The “true” gradient is taken as the mean $g = \frac{1}{N}\sum_i g_i$. If we assume that the sparse codes recovered with and without $A_i$ are the same, we have for each sample $\nabla_D F({Z_A}_i (D), D) = A_i^T A_i g_i$. Thus, $\nabla_D  F(Z_A(D), D) = \frac{1}{N}\sum_i A_i^T A_i g_i$.
> Let u be such that $u$ is in the intersection of $\ker(A_i)$, supposed non empty, then if $D_S$ is solution of $\nabla_D  F(Z_A(D), D) = 0$, then $D_S + u$ is also solution.
> In the paper, we gave the simpler version where all g_i are equal to make it easier to visualize because of the factorization, but taking multiple samples doesn't change the conclusion. We will update the paper to make this clearer.
>
> **Other Minor points:**
>
> * Dictionary visualization - We added a figure in the Appendix to compare dictionaries learned in inpainting in deblurring. As explained in the paper, as high frequencies are put to 0 in the case of deblurring;, the atoms appear blurry in our figure.
>
> * Figure positions - We will do our best regarding Latex management of the floating figures and the limited space.
>
> * Shared y-axis in figure - We tried doing this, but since the subplots have different scales, it makes visual interpretation harder.

---

### Official Review · Reviewer_zjwn · 2022-10-27

**Confidence:** 3
**Correctness:** 2
**Technical Novelty And Significance:** 2
**Empirical Novelty And Significance:** 2
**Recommendation:** 3

**Clarity, Quality, Novelty And Reproducibility:**

I am quite confused about what exactly the authors are trying to claim in this paper. They analyze some cases but never really connect the conclusions together in a convincing and straightforward way. I would say the novelty of the analysis presented is modest at best. They do provide code for reproducibility.



**Strength And Weaknesses:**

-  I am not quite sure what the contributions of the paper are. The implications drawn from Prop 2.1 and 2.2 have been known for a while in inverse problems.
- Similarly in the experiments, I do not think that SNR dropping in the kernel space while mostly stable in the range space is a novel observation; it is expected behaviour.
- I don’t quite understand Eq 9, shouldn’t A only be applied to f_\theta(Y_i)? In Eq 8, f_theta was taking in terms from the space of images (i.e. X space) while in 9 f_theta takes in measurements (Y-space)? Can the authors clarify how 8 leads to 9?
- I do not find the WSS justification for convolutions convincing? Why is it ok for X  (an image) to be considered a WSS process?  I think the reasoning could have been given without any WSS assumption. Consider any image x, if we apply a random mask with masking probability p, the expected norm of the masked image should automatically be rho*\|x\|. What am I missing here?
- “Natural images are stable enough to allow convolution-based algorithms to learn from all frequencies” – what does it mean for natural images to be stable? Also DIP or any other network has more than convolutions. In fact non-linearities are essential for a network’s performance. What is the claim here?
- In Fig 3 and 6, please also include the image over which the experiment was run.
- I also don’t understand what the authors mean to say when they claim that convolutions won’t work on deblurring. We do have sharpening filters that work to sharpen slightly blurred images. Is the claim that CNNs when trained DIP-style can’t deblur?


**Summary Of The Paper:**

The paper tries to answer when prior learning can help in inverse problems using unsupervised ML methods. I don’t think this is an easy question to answer yet and understandably the paper considers a few simpler cases.  The paper starts off with traditional dictionary learning methods and shows that one can only learn atoms in the range of the operator. Next, they discuss convolutional dictionary learning and finally provide some empirical observations about unsupervised learning methods.

**Summary Of The Review:**

I think the paper lacks few clear novel ideas that it wants to bring to the community. A lot of known conclusions are presented but unfortunately do not lead to something concrete or novel that the community can benefit from. I cannot recommend acceptance at this stage.

---

> ### Author Response · Authors · 2022-11-15
> **Answer to reviewer zjwn**
>
> We thank the reviewer for the feedback.
>
> **Novelty -** Propositions 1 and 2 appear essential to state that "seeing the whole space" is necessary to learn the dictionary in a general sparse coding context. Even if this result can be seen as a consequence of Tachella et al. (2022), this specific formulation may need to be clarified for the community. Moreover, we provide simple and constructive proof that may help understand these results and implications.
> Please see the general answer we made to all reviewers concerning the novelty of the work and the implications.
>
> **Drop of SNR -** The observation is not the drop of SNR between the kernel space and the range space but the drop difference between inpainting and deblurring.
>
> **Multiple application of A in Eq (9) -** In the case of inpainting, both formulations are equivalent as AA = A. This changes in the case of deblurring with gaussian filters. The motivation behind using the loss in the paper and not this loss is that the operator may change the scale of the data, and the network won’t only learn to denoise but also to perform a rescaling of the data. We add the operator before Y_i to be sure that the “observations” Af(Y_i’) and AY_i have the same scale. For information, we tried the loss proposed by the reviewer at first, and it led to poor results experimentally.
> 8 doesn’t lead to 9. Plug and play methods described in 8 use a denoiser that can be trained on data. The goal was to verify whether training the denoiser in the range space of the operator was enough to solve the inverse problem, leading to the loss proposed in 9.
>
> **WSS and convolution -** We wanted to emphasize the behavior of the mask on the spectrum of a simple signal. When a mask is applied to a WSS, the spectrum of the observation is proportional to the original spectrum at each frequency. The WSS assumption is required to define the density spectrum properly. This result is stronger than just considering the whole power of the signal, as mentioned by the reviewer.
> Moreover, even though images are not stationary, the patches from the image can be supposed to be WSS. Convolutional sparse coding allows spotting repetitions and mixings of these stationary patches.
>
> **Natural images have enough redundancy -** The claim is that natural images are somehow stationary at a small scale, as explained above. Coupled with the fact that all frequencies are observed at a small scale in inpainting, this explains why convolutions help in this context from a signal processing point of view. Of course, non-linearities are essential to the networks or dictionary learning. However, the experiments show that the same networks or methods applied to deblurring do not work, even with non-linearities. This highlights how essential it is to introduce proper weak prior knowledge to the reconstruction method.
>
> **Convolutions won’t work on deblurring: clarifications -** In the last paragraph of subsection 3.2, we say that convolutions are insufficient to deblur. Adding stronger prior information is necessary to compensate for the lack of information in high frequencies. Sharpening filters are a type of prior information that adds high frequencies to the output. We will refer to this type of non-linearity in the paper. This is not specific to Deep Image Prior but applies to Plug and Play and Convolutional Dictionary Learning.

---

### Official Review · Reviewer_SBkU · 2022-10-31

**Confidence:** 4
**Correctness:** 3
**Technical Novelty And Significance:** 2
**Empirical Novelty And Significance:** 2
**Recommendation:** 3

**Clarity, Quality, Novelty And Reproducibility:**


**Clarity:** Overall, the paper was easy to read. There were some notational issues and questions regarding derivations that I was looking to clarify.

- I am confused by the notation at the bottom of page 3. Should the objective functional be denoted by $F(Z_A(D), D)$ to be consistent with the gradient derivations on the following page?
- Also, is equation (3) the true “gradient” of the objective? We should have sub gradients due to the ell1 norm penalty on $Z_{A_i}(D)$. Moreover, this quantity itself depends on $D$, so I think the full sub gradient should be more involved.
- In equation (9), should each term in the loss be $||Af_{\theta}(Y_i’) - Y_i||_2^2$? Otherwise, I am not sure I understand why one should apply the degradation operator $A$ to the already degraded measurements $Y_i = AX_i + \epsilon_i$.

**Novelty and significance:**

I think that while the topic tackled is timely, the results are not very significant. The results are very similar to previous results in a slightly different setting. The experiments make some interesting comments on when priors can be learned (namely, when the structure of the priors compensate for the type of information lost in the degradation operator). However, I feel as though there’s a disconnect between the presented theory and the experiments. In particular, the experiments seem to make a novel claim about what situations allow for prior learning, but the theory does not tackle these cases and, instead, shows results similar to previous results but in a particular setting (namely, dictionary learning). The particular results I am referencing are those of Tachella et al. (2022), that show recovery is not possible in the kernel of the measurement operator (but is possible with access to examples from several forward models).

Additionally, some of the experimental comments being made in this paper have been made before in previous works. For example, the Deep Image Prior paper effectively shows that convolutional structure aids in recovering signals from noisy measurements with only access to an implicit prior. This paper shows this phenomenon also holds in models based on Convolutional Dictionary Learning and other frameworks, but the punchline is effectively unchanged. I think further theory along these lines would be needed to elevate the contributions of this paper.

**General comments:**
- One question I had while reading the main text was what the role or influence the number of examples $N$ played in the recovery error/success rate? This information can be found in the appendix (namely, that the number of samples was kept high so that this was not a factor in the results), but it would be good to note this in the main text.


**Strength And Weaknesses:**


**Strengths:**
- The paper studies an important and challenging problem, which is to identify when and why its possible to learn priors directly from corrupted data.
- The experimental make some interesting comments on when priors can be learned, which can happen when information is introduced into the prior model that can compensate for the type of degradation in the forward model.

**Weaknesses:**
- There is a lack of novelty in the theory presented in this paper. The results presented here are very similar in spirit to previous work. Please see the Novelty and significance section below.


**Summary Of The Paper:**

This paper considers learning priors in the case when one only has access to corrupted measurements of signals, rather than clean signals themselves. The authors first provide an analysis of learning a sparsifying dictionary when only given compressed measurements. It is shown that if only one measurement operator is used, then information in the kernel of the measurement operator cannot be recovered. The situation changes, however, when multiple measurement matrices are used which are “diverse” enough for information to be learned. The authors then experimentally demonstrate that recovery with only corrupted data is possible when convolutional structure is used as a prior in certain problems, namely inpainting. This is shown to not be the case, however, in deconvolution.

**Summary Of The Review:**

I think that the paper is tackling an interesting problem and showcases some promising/interesting interesting experimental observations. However, in its current form, I think that it lacks novelty/significance compared to prior work in both the theory and experiments.

---

> ### Author Response · Authors · 2022-11-15
> **Answer to reviewer SBkU**
>
> We thank the reviewer for the feedback.
>
> **Novelty** We refer the reviewer to our general answer regarding the novelty and goal of our work, particularly that it complements the one from Tachella et al. (2022). While our results are not surprising, we feel that these issues and claims are somehow overlooked by part of the community and need practical clarifications and illustrations. This paper will encourage the community to research ways to impose weak prior for unsupervised inverse problem resolution, either with equivariance as proposed in Tachella et al. (2022) or with domain knowledge such as convolution and non-linearities for some signal processing tasks.
>
> **Notation page 3  -** Absolutely, we corrected the paper appropriately.
>
> **True gradient in Eq (3) -** This objective is differentiable over the dictionary D. This is a consequence of the Danskin’s theorem (see Mairal et al. 09 for a detailed explanation). Therefore, there is no need for subgradients.
>
> **Multiple application of A in Eq (9) -** In the case of inpainting, both formulations are equivalent as AA = A. This changes in the case of deblurring with gaussian filters. The motivation behind using the loss in the paper and not this loss is that the operator may change the scale of the data, and the network won’t only learn to denoise but also to perform a rescaling of the data. We add the operator before $Y_i$ to be sure that the “observations” $Af(Y_i’)$ and $AY_i$ have the same scale. Note that we tried the loss proposed by the reviewer at first, and it led to poor experimental results.

---

### Author Response · Authors · 2022-11-15
**General answer on novelty**

We thank the reviewers for their many comments that will help us improve the quality of the paper. All reviewers acknowledge that the paper tackles a challenging, interesting, and timely problem. Yet, some reviewers seem to have missed our work's goal, which we clarify here and in our revised manuscript.

Our primary goal is to see whether it is possible to apply appealing new methods like PnP, DIP, or CDL developed for natural images to challenging inverse problems where no ground truth is available, such as in neuroscience or astrophysics. To demonstrate the current practical limitations of these methods, we apply them to imaging problems where performances can easily be evaluated. From our point of view, the two significant contributions are the following:

**Dictionary Learning -** What is known in the literature is that nothing can be learned in the kernel space of the measurement operator and that using several operators allows for mitigating that issue (Tachella et al., 2022). In other words, "seeing the whole space" is a necessary condition to learn a prior from the data. However, up to our knowledge, the way the operator(s) impact(s) the prior learning process, even in the range space, and in particular to what extent the previous condition is sufficient, has yet to be studied. In the case of dictionary learning, we emphasize that "seeing the whole space" might not be enough to learn a proper prior (fig 1, 2, 3). To demonstrate that, we first return to the case where only one single operator exists and show that the dictionary can't be appropriately learned even in the range space due to the dimension reduction. Then, we study the case where multiple operators degrade the data and show that the problem is made harder than it would be in the case of denoising and that it is sometimes unfeasible even with access to the whole space. Hence, "seeing the whole space" is indeed not sufficient. These results are useful for the inverse problem community, as they complement the ones from Tachella et al. (2022).

**Convolutions for inverse problems -** The goal is to study the behavior of methods heavily relying on convolutions in cases where they work well and in cases where they fail because the prior is too weak. In the first case, we chose the example of inpainting, and we claim that this problem is made relatively easy by the fact that all signal frequencies are preserved. Thus, a broad range of methods based on convolutions will perform well. We presented three examples, namely PnP, DIP, and CDL. The difference with previous works is that we train the prior "as is" in the range space without relying on data augmentation techniques or equivariance to show that the task is feasible. Of course, data augmentation or equivariance may improve performance. In the second case, we experimentally confirm Prop 2.2: i.e., convolutions are insufficient in deblurring in an unsupervised setting. To show that the difficulty is deeper than the "unsupervised setting," we also add a study on what happens when the denoiser is trained in a self-supervised setting (dictionary and denoiser learned on ground truth data). We show that stronger prior information is necessary to link low and high frequencies, even in this simpler case.

In addition to the new contributions mentioned above, the experiments we present in the paper also provide complementary and interesting information to the theoretical study of Tachella et al. (2022). We believe these conclusions are of high importance for  the inverse problem community, and that our experiments will broaden their impact. Finally, our paper opens interesting research directions that would empower methods like PnP, DIP, and CDL for inverse problems.

---

> ### Comment · Reviewer_ZBdm · 2022-11-25
> **Reviewer comment after reading the authors' response**
>
> I thank the authors for their response. I see the paper as a practical analysis of learning priors and their relation to single/multiple operators and their structures. The analysis of this paper is valuable to the inverse problem community, especially those dealing with unsupervised problems. Although there still seems to be a novelty concern from other reviewers, I find this paper a useful complement to (Tachella et al., 2022), and recommend its acceptance (I keep my rating the same).

---

### Author Response · Authors · 2022-11-18
**End of rebutal period**

We thank the reviewers again. We hope the novelties have been clarified and the appreciations will be revised accordingly, in particular the correctness of the paper. The paper seems technically correct and the claims are supported. We kindly ask reviewers SBkU and zjwn to indicate which parts still appear incorrect or not well supported, if any. We are, of course, available to bring any more clarification if asked. Thank you again for the valuable comments, which help to clarify the paper.

---

### Decision · Program_Chairs · 2023-01-20

**Decision:**

Reject

**Justification For Why Not Higher Score:**

As mentioned in several reviews and the summary above, the contribution of this paper is somewhat too incremental to merit acceptance.

**Justification For Why Not Lower Score:**

N/A

**Metareview: Summary, Strengths And Weaknesses:**

The paper studies the problem of dictionary learning in the setting where the sampled training data is only accessible via incomplete linear measurement operators. In the single-operator setting, the authors show that learning the 'true' dictionary is impossible. However, if multiple measurement operators are available, the problem becomes less challenging (but still, difficult in the presence of noise). To remedy this, the authors suggest using regularization schemes (or 'priors') such as plug-n-play, or deep-image priors, or convolutions; these improve performance in some cases.

The paper received split scores. Some reviewers appreciated the insights provided by the suggested regularization schemes. Other reviewers pointed out issues with writing clarity, justification for certain assumptions/claims, and a lack of sufficient explanation of how the results would generalize to other, more practically relevant settings. Importantly, all reviewers raised a common concern about novelty; similar observations have been made several times in the past (and most recently, in the exact same setting of dictionary learning from incomplete observations, by Tachella et al. '22). Therefore, the contributions of this paper -- as presented -- are modestly incremental, and an addendum to this earlier paper at best.

Recommendation: reject.

**Summary Of Ac-Reviewer Meeting:**

This was not a borderline paper.